# *IFN-β* Overexpressing Adipose-Derived Mesenchymal Stem Cells Mitigate Alcohol-Induced Liver Damage and Gut Permeability

**DOI:** 10.3390/ijms25158509

**Published:** 2024-08-04

**Authors:** Soonjae Hwang, Young Woo Eom, Seong Hee Kang, Soon Koo Baik, Moon Young Kim

**Affiliations:** 1Department of Biochemistry, Lee Gil Ya Cancer and Diabetes Institute, College of Medicine, Gachon University, Incheon 21999, Republic of Korea; soonjae@gachon.ac.kr; 2Regeneration Medicine Research Center, Wonju College of Medicine, Yonsei University, Wonju 26426, Gangwon-do, Republic of Korea; yweom@yonsei.ac.kr (Y.W.E.); baiksk@yonsei.ac.kr (S.K.B.); 3Cell Therapy and Tissue Engineering Center, Wonju College of Medicine, Yonsei University, Wonju 26426, Gangwon-do, Republic of Korea; 4Department of Internal Medicine, College of Medicine, Korea University, Seoul 02841, Republic of Korea; shkang0114@gmail.com; 5Department of Internal Medicine, Wonju College of Medicine, Yonsei University, Wonju 26426, Gangwon-do, Republic of Korea

**Keywords:** mesenchymal stem cell, binge alcohol, gut leakiness, *IFN-β*, HGF

## Abstract

Alcoholic liver disease (ALD) is a form of hepatic inflammation. ALD is mediated by gut leakiness. This study evaluates the anti-inflammatory effects of ASCs overexpressing interferon-beta (ASC-IFN-β) on binge alcohol-induced liver injury and intestinal permeability. In vitro, ASCs were transfected with a non-viral vector carrying the human *IFN-β* gene, which promoted hepatocyte growth factor (HGF) secretion in the cells. To assess the potential effects of ASC-IFN-β, C57BL/6 mice were treated with three oral doses of binge alcohol and were administered intraperitoneal injections of ASC-IFN-β. Mice treated with binge alcohol and administered ASC-IFN-β showed reduced liver injury and inflammation compared to those administered a control ASC. Analysis of intestinal tissue from ethanol-treated mice administered ASC-IFN-β also indicated decreased inflammation. Additionally, fecal albumin, blood endotoxin, and bacterial colony levels were reduced, indicating less gut leakiness in the binge alcohol-exposed mice. Treatment with HGF, but not IFN-β or TRAIL, mitigated the ethanol-induced down-regulation of cell death and permeability in Caco-2 cells. These results demonstrate that ASCs transfected with a non-viral vector to induce *IFN-β* overexpression have protective effects against binge alcohol-mediated liver injury and gut leakiness via HGF.

## 1. Introduction

Alcoholic liver disease (ALD) is one of the most prevalent liver diseases worldwide [1]. Excessive alcohol consumption can result in liver cirrhosis attributed to fatty liver and steatohepatitis. Approximately 50% of cirrhosis-related deaths are associated with alcohol-induced liver diseases [2]. Excessive alcohol consumption can also cause significant damage to other major organs, including the gut, leading to leakiness [3], contributing to increased endotoxin levels and inflammatory tissue damage to the liver [4]. In non-clinical studies, it has been found that alcohol-induced liver damage causes with intestinal damage, thereby inducing intestinal inflammation and leakiness [5]. Intestinal bacteria or bacterial products translocate into the liver via a leaky barrier [5].

Adipose-derived mesenchymal stem cells (ASCs) are easy to obtain clinically [6], and they exhibit therapeutic effects in several inflammatory disease models [6,7,8,9], including hepatitis mouse models. Thus, research has been conducted in the direction of enhancing the therapeutic efficacy of ASCs [10,11]. Enhancing the function of ASCs using a viral vector shows better therapeutic efficacy than conventional ASCs in animal models with inflammatory diseases [12,13,14]. However, ASCs engineered using viral-vector-mediated transduction are challenging to use clinically or obtain official approval as a drug due to the risk of viral vector-induced immune response [15,16]. Therefore, research using non-viral vectors is being attempted to avoid non-specific immune responses in vivo.

A previous study demonstrated that cultivating high-density human ASCs induces interferon-beta (IFN-β secretion [17,18]). Secreted IFN-β promotes tumor necrosis factor-related apoptosis-inducing ligand (TRAIL) production in human ASCs [17]. TRAIL induces antitumor effects via cell cycle arrest or apoptosis based on the type of tumor cells [17,18,19]. Moreover, IFN-β treatment reportedly reduces the expression of interleukin (IL)-1β and tumor necrosis factor (TNF)-α [20,21], which play a major role in the pathogenesis of ALD [22] and leaky gut [23,24]. A recent study showed that mouse IFN-β induces mouse hepatocyte growth factor (HGF) via signal transducer and activator of transcription 1 (STAT1) pathway in mouse mesenchymal stem cells [25]. IFN-β also stimulated HGF secretion in human peripheral blood mononuclear cells [26]. HGF treatment accelerates wound healing in major organs of humans and mice [27,28,29]. However, no studies have used human ASCs transfected with a non-viral vector carrying the human *IFN-β* gene to treat ALD.

In the current study, a non-viral vector harboring the *IFN-β* gene was constructed and transfected into ASCs (resulting in ASC-IFN-β). This study aimed to evaluate the anti-inflammatory potential of ASC-IFN-β in a murine experimental model of binge alcohol consumption. We found that injection of ASC-IFN-β decreased alcohol-induced liver injury by preventing gut leakiness via HGF secreted by IFN-β-treated ASCs in part. Treatment with HGF suppresses alcohol-induced cell death and disruption of junctional complexes in intestinal epithelial cells in vitro. Our results indicate that overexpression of IFN-β in ASCs using a non-viral vector may further improve the therapeutic effects of ASCs on alcoholic liver diseases via HGF. 

## 2. Results

### 2.1. Overexpressing Human ASCs with IFN-β Induces HGF Secretion

It was hypothesized that engineering human MSCs to overexpress IFN-β induces HGF expression. Before testing the hypothesis, firstly, human ASCs were treated with IFN-β in a dose-dependent manner (100 to 1000 units/mL) for 24 h. It was confirmed that human IFN-β treatment (100 to 500 units/mL) induced human HGF expression in human ASCs (Figure 1A,B), which then prompted us to transfect human ASCs with IFN-β using a non-viral vector. Similarly, overexpressing IFN-β in human ASCs induced HGF expression and secretion (Figure 1C,D).

### 2.2. Treatment of ASCs Expressing IFN-β Decreases Binge Alcohol-Induced Liver Damage in Mice

To determine the anti-inflammatory effects of human ASCs overexpressing IFN-β on alcohol-induced liver injury, C57BL/6 mice were treated with binge alcohol (three oral doses of 6 g/kg/dose at 12 h intervals). The binge alcohol-treated mice were administered intraperitoneally human ASCs overexpressing IFN-β (ASC-IFN-β) immediately after the second treatment of alcohol (Figure 2A). After the third treatment of alcohol, the mice were sacrificed, followed by an analysis of body weight. Alcohol-treated mice given ASC-IFN-β showed a significantly less decrease in body weight compared with mice given alcohol alone (Figure 2B). Next, we hypothesized that ASC-IFN-β decreased alcohol-induced liver inflammation. To test this hypothesis, histological staining of liver tissues was performed. Histology showed that the treatment of ASC-IFN-β but not ASC-vec decreased hepatic inflammation and steatosis in alcohol-exposed mice (Figure 2C). Consistent with these results, liver weight, AST, and ALT were reduced in alcohol-exposed mice treated with ASC-IFN-β compared with the group given ASC-vec (Figure 2D–F). The expressions of TNF-α, IL-1β and iNOS were also reduced in the liver tissue of alcohol-exposed mice administered ASC-IFN-β (Figure 2G–I). These results indicate that ASC-IFN-β treatment decreases liver injury and inflammation in binge alcohol-exposed mice.

### 2.3. Treatment of ASCs Expressing IFN-β Reduces Binge Alcohol-Induced Gut Injury and Leakiness

To determine if ASC-IFN-β reduces alcohol-induced intestinal damage, histologic assessment was performed on the intestinal tissue. The results of histology showed that abnormal shapes (increased villi length and decreased crypt length) of intestines were frequently observed in alcohol-exposed mice compared to sham (Figure 3A–D). Furthermore, elevated expression of intestinal TNF-α and IL-1β and iNOS of alcohol-exposed mice was reduced in alcohol-exposed mice treated with ASC-IFN-β (Figure 3E–G). As intestinal inflammation decreased in alcohol-exposed mice treated with ASC-IFN-β, next, we evaluated gut leakiness (fecal albumin, plasma endotoxin, and the colony-forming unit (CFU) of *E. coli* present in liver tissue). Consistently, treatment of ASC-IFN-β reduced the levels of the gut leakiness markers in the alcohol-exposed mice, while treatment of ASC-vec showed comparable levels of gut leakiness markers with those in mice treated with alcohol alone (Figure 3H–J). These results show that treatment with ASC-IFN-β reduced alcohol-induced gut inflammation and leakiness in binge alcohol-exposed mice.

### 2.4. HGF Reduces Ethanol-Induced Cell Death and Barrier Disruption in Intestinal Epithelial Cells

To investigate the protective mechanism of ASC-IFN-β, firstly, Caco-2 intestinal epithelial cells were treated with 40 mM ethanol for 24 h, and then caspase 3 activity, a cell death marker, was evaluated. The results showed that ethanol-exposed Caco-2 showed elevated caspase 3 activity compared with that in the control group (Figure 4A). Ethanol-exposed Caco-2 treated with IFN-β or TRAIL showed comparable caspase 3 activity with the group treated with ethanol alone (Figure 4A,B). In contrast, Caco-2 cells co-treated with 40 mM ethanol and HGF (10 to 40 ng/mL) for 24 h showed reduced caspase 3 activity compared with that in the group treated with 40 mM ethanol alone (Figure 4C). These results show that HGF treatment decreases ethanol-mediated cell death. Similarly, HGF treatment prevented ethanol-induced barrier permeability in Caco-2 cells (Figure 4D,E). Single treatment of IFN-β, TRAIL, or HGF does not affect caspase 3 activity and barrier permeability in control Caco-2 cells (Figure 4A–E). These results support the protective roles of HGF from ASC-IFN-β in the alleviation of ethanol-induced liver injury and gut leakiness.

## 3. Discussion

Our findings indicate that treatment with ASC-IFN-β effectively reduces ethanol-induced hepatic inflammation via HGF-mediated protection of gut leakiness. Although clinical studies have revealed diverse outcomes regarding the therapeutic efficacy of ASCs on ALD treatment, ASCs are still regarded as potential pharmaceuticals for stem cell-based liver disease therapies. Thus, adult stem cell research has focused on developing effective methods for improving the therapeutic potential of ASCs. In the present study, we sought to develop a strategy to enhance the therapeutic efficacy of ASCs in vivo using a commercial non-viral vector to induce the overexpression of *IFN-β* in ASCs, which were then intraperitoneally injected into binge alcohol-exposed mice.

ASCs cultured at high densities secrete TRAIL via *IFN-β* [17]. As TRAIL is known to cause apoptosis in several cancer cells [30,31,32], it can be used to suppress ASC-induced tumorigenesis; moreover, treatment with IFN-β has also been reported to suppress cancer in breast [19] and hepatic [18] cancer cells. Therefore, therapeutic strategies using ASC-IFN-β have been studied in rodent models to suppress tumors [33,34].

Our results demonstrated that intraperitoneal injection of ASC-IFN-β was sufficient to improve binge alcohol-induced liver injury and gut leakiness via *IFN-β*-induced HGF secretion. Specifically, treatment with HGF, but not IFN-β or TRAIL, directly suppressed the disruption of the intestinal barrier in the Caco-2 cells. Therefore, we believe that HGF secreted by ASC-IFN-β has protective effects on damaged gut barrier and epithelial permeability. To our knowledge, this is the first study in which intraperitoneal injection of human ASC-IFN-β-induced HGF was tested for binge alcohol-induced liver injury and gut leakiness. IFN-β and TRAIL did not prevent ethanol-induced destruction of the epithelial barrier. However, chronic alcohol consumption may inhibit hepatic fibrosis and carcinogenesis in patients with liver cirrhosis or cancer.

Cho et al. [5] showed that chronic fructose-mediated steatosis and liver inflammation are promoted by disruption of the gut barrier. Fructose-induced leaky gut also induces intestinal bacteria or bacterial endotoxins into the liver tissue via the blood [35]. Other studies have indicated that oral treatment with therapeutic materials, such as prebiotics, probiotics, and natural products, showed protective effects against steatosis or liver inflammation by improving the gut environment, thereby decreasing gut leakiness [36,37]. Engineered ASCs secreting HGF in response to *IFN-β* might be effective in fructose-mediated liver diseases by improving the intestinal barrier. Further research is warranted to investigate whether engineered ASCs are also effective in patients with nonalcoholic liver fatty disease or metabolic syndrome characterized by leaky gut symptoms.

As lentiviral vectors usually carry human telomerase reverse-transcriptase and c-Myc reprogramming factors [38], the use of genetically engineered ASCs by lentiviral vectors may contribute to tumorigenicity in clinical applications. To exclude the possibility of tumorigenesis, we used a commercial non-viral vector to overexpress *INF-β*-mediated HGF. The use of a non-viral vector results in a relatively low systemic immune response when compared to that of a lentiviral vector. ASCs themselves are home to the site of inflammation; hence, they can also induce transduction in other normal cells through viral vectors in normal tissues. Therefore, they have the advantage of countering safety concerns.

A strategy for directly overexpressing HGF in bone marrow–mesenchymal stem cells or ASCs has been widely used [34,39,40]. HGF overexpression in this manner increases the therapeutic efficacy of mesenchymal stem cells; however, additional secretion of *IFN-β* or TRAIL through HGF is not possible. Nevertheless, overexpressing *IFN-β* in ASC can induce TRAIL and HGF through the IFN-β/STAT pathway. Therefore, engineered ASCs that migrate to the intestinal and liver tissues may have synergistic effects that inhibit and regulate local inflammation and promote the repair of damaged tissues through a combination of the three cytokines secreted by the engineered ASC.

It is challenging for a therapeutic approach using ASCs with enhanced functions to produce the same recovery effect as a liver transplant in a healthy individual. Accordingly, in addition to the functionally enhanced ASCs, Fecal Microbiota Transplantation (FMT) may be considered a therapeutic alternative for patients with irreversible alcoholic cirrhosis. FMT has demonstrated benefits in enhancing resistance to microbial infections and improving metabolic syndrome indicators in clinical studies [41]. Given the confirmed beneficial effects of FMT [41], it is believed that this approach can be applied not only to alcoholic liver disease but also to other intestinal diseases via combined administration with functionally enhanced ASCs. Future clinical studies evaluating the effects of this combination therapy are warranted.

In other words, our study demonstrated that a non-viral vector-engineered ASC-IFN-β was capable of inducing HGF secretion, which decreased ethanol-induced liver injury and gut leakiness by suppressing epithelial barrier permeability. Thus, a strategy for *IFN-β*-mediated HGF secretion by non-viral vectors could be considered a safe and feasible approach for treating ALD. Further studies are warranted to evaluate the application of ASC-IFN-β in the treatment of patients with liver fibrosis for stem cell therapy. 

## 4. Materials and Methods

### 4.1. ASC Culture and Characterization

Collagenase (Roche, Munich, France) was used in primary ASCs to separate cells from fat tissues, and three healthy donors provided informed consent. The adipose tissue samples were extensively washed with Hank’s Balanced Salt Solution (HBSS; Sigma-Aldrich, St. Louis, MO, USA) to remove any unwanted materials. They were then treated with a dilute collagenase solution (0.025% in HBSS) for an hour at 37 °C under gentle agitation. The remaining debris from the adipose tissue was removed by filtration through a 100 μm mesh filter (SPL Life Science, Pocheon, Republic of Korea). In total, 5 million mononuclear cells were seeded in 100 mm culture dishes containing DMEM Sigma-(Aldrich, St. Louis, MO, USA) supplemented with 10% fetal bovine serum and antibiotics. After 2 days, the culture medium was changed to remove any non-adherent cells. The cell culture medium was then changed twice weekly, and the cells were passaged using trypsin/EDTA (Sigma-Aldrich, St. Louis, MO, USA) once they reached 80% confluency. For experimental studies, the cells were seeded at a density of 40,000 cells per square centimeter and cultured for the desired duration. The conditioned medium was collected, filtered, and stored at −80 °C for future use. The adipose-derived stem cells (ASCs) were maintained at 37 °C and 5% CO_2_, with the culture medium replaced every 3–4 days. When the cells reached 70–80% confluency, they were detached using trypsin and passaged. All the protocols were reviewed and approved by the Institutional Review Board of Yonsei University Wonju College of Medicine (CR319098) and the Institutional Biosafety Committee (19-6 and 19-7) of Yonsei University Wonju College of Medicine. Cryopreserved ASCs were seeded in 6-well plates, and the culture medium containing 10% fetal bovine serum (FBS) was changed once every 2 days. Differentiation potential and surface cell antigen analysis of ASCs were performed as described as in a previous study [18].

### 4.2. Transfection of Non-Viral Vector to Carry Plasmid Containing the IFN-β Gene in the ASCs

Mesenchymal stem cells were plated onto a 100 mm dish for 24 h, and transfection was performed at 70–80% confluence. The *IFN-β* DNAs were transfected into the ASCs via X-tremeGENE™ Transfection Reagents (Roche, Munich, France). Transfection reagents (2 μL) and DNA (4 μg) were diluted in 1 mL of a serum-free medium, followed by equilibration at room temperature for 30 min after mixing. The transfection reagents–DNA complex was introduced into the ASC culture media and incubated for 48 h. Subsequently, the ASCs underwent a washing step using phosphate-buffered solution (PBS), following which the medium was substituted with Dulbecco’s Modified Eagle Medium (DMEM) enriched with 10% fetal bovine serum (FBS).

### 4.3. Evaluation of the Caspase Activity and Barrier Function in Caco-2 Cells

Caspase activity was assessed using Caspase-Glo 3/7 assay kits (Promega, Madison, WI, USA) following the provided manual. The human colonic Caco-2 cells were cultured on polycarbonate membrane Transwell inserts with a surface area of 0.33 cm^2^ and a pore size of 0.4 μm (SPL Life Science, Pocheon, Republic of Korea). Following a one-day seeding period, the cells were maintained in fresh DMEM supplemented with 10% FBS. The culture medium was refreshed three times weekly. Upon achieving a trans-epithelial electrical resistance (TEER) exceeding 350 Ω/cm^2^, the intestinal barriers composed of Caco-2 cells were exposed to ethanol alone or in combination with IFN-β, TRAIL, or HGF for 24 h. After treatment, TEER was assessed using a Millicell Electrical Resistance System (ERS) meter (Millipore Corporation, Bedford, MA, USA). Data were analyzed from four inserts per treatment in three separate experiments and presented as a percentage relative to the initial TEER value before ethanol exposure. After TEER measurements, 50 μg/μL fluorescein isothiocyanate-labeled 4-kDa dextran (FITC-D4; Sigma-Aldrich, St. Louis, MO, USA) was introduced to the apical side of the upper chamber of the Caco-2 cell-seeded Transwell inserts. Following a one-hour incubation with FITC-D4, the intestinal permeability was evaluated using spectrophotometry.

### 4.4. ELISA

Culture supernatants of ASCs were collected after *IFN-β* transfection at different time points (0, 24, and 48 h). The collected culture supernatants were centrifuged and frozen before the enzyme-linked immunosorbent assay (ELISA). Culture supernatants were analyzed for HGF using a commercial ELISA kit (R&D Systems, Minneapolis, MN, USA), following the manufacturer’s instructions.

### 4.5. Animal Treatments

Eight-to-ten-week-old female C57BL/6 mice (Raon-Bio Company, Yongin, Republic of Korea) were co-housed for acclimatization in the vivarium for two weeks before randomization and then maintained under controlled lighting (12 h light/dark cycle) with food and water provided ad libitum. The mice were administered three oral doses of binge alcohol (6 g/kg/dose) or dextrose (control) at 12 h intervals and euthanized 1 h after the final binge alcohol injection. The mice were administered a single intraperitoneal injection of 2.0 × 10^6^ ASCs/200 µL PBS on day 2 after receiving a second binge alcohol dose. The experimental procedures of housing and breeding mice were reviewed and approved by the Institutional Animal Care and Use Committee of Yonsei University at Wonju (YWCI-201909-013-01 and YWC-200305-1). 

### 4.6. Histology

The livers and ilea of the mice were excised and fixed in 10% formalin. The paraffin blocks were sectioned (8 μm) using a rotary microtome (Leica, Wetzlar, Germany). Tissue sections were heated on slides to eliminate water between the paraffin sections and the slide surface, which enhanced the stickiness of the sections to the slides. Slides were serially stained with hematoxylin and eosin (H&E) as previously described [42]. H&E staining was performed under an optical microscope (Leica, Wetzlar, Germany). Histological images were processed using Adobe Photoshop 7.0 (Adobe, San Jose, CA, USA) and Leica software 5.2.2 (Leica, Wetzlar, Germany). The assessment of ileal inflammation was carried out via H&E staining, considering factors such as local inflammation severity, mucosal injury extent, tissue regeneration, and crypt damage. The cumulative scores for all parameters were computed. The evaluation of ileum inflammation was categorized as follows: 0 for normal, 1 for a mild increase in immune cells with no colonic epithelial alterations, 2 for a moderate increase in immune cells with mild epithelial proliferation, and 3 for a severe increase in immune cells with aberrant epithelial proliferation leading to extensive loss of villi/crypt architecture. Inflammation scores and villus and crypt lengths were determined as the median of measurements from four sections per slide for each mouse.

### 4.7. Endotoxin Measurement

Blood was isolated from the mice via cardiac puncture. After centrifugation of blood (3000 rpm, 20 min, 4 °C) in a heparin tube (BD Biosciences, San Jose, CA, USA), the supernatant of the blood was isolated and frozen until use. The serum of the mice was examined for endotoxin levels using an endotoxin quantitation kit (Thermo Fisher Scientific, Waltham, MA, USA) according to the manufacturer’s instructions.

### 4.8. Albumin ELISA

Mouse feces were gathered in sterile microfuge tubes and placed in a sample buffer for dilution. The levels of albumin in the feces were assessed using an ELISA kit from Abcam in Cambridge, UK. The experimental protocols adhered to the guidelines provided by the manufacturer, and the optical density of the ELISA plates was measured using a microplate reader from BioTek Instruments (Winooski, VT, USA).

### 4.9. Serum Alanine and Aspartate Transaminase Measurements

Mouse sera were transferred to the Raon-Bio Company (Yongin, Republic of Korea) to analyze alanine transaminase (ALT) and aspartate transaminase (AST) levels. All the samples were diluted with PBS. The median was considered the representative value for each mouse. Hemolytic sera were excluded from the statistical analysis.

### 4.10. Quantitative Reverse-Transcription Polymerase Chain Reaction

RNA was isolated from liver and ileum tissues using TRIZOL reagent (Thermo Fisher Carlsbad, CA, USA) according to the manufacturer’s guidelines. The RNA concentration was quantified and employed for complementary DNA synthesis. Transcript levels for individual gene targets were determined using the QuantStudio™ Real-Time PCR System (Applied Biosystems, Waltham, MA, USA). qRT-PCR data were normalized to *GAPDH* expression, and the relative amount was calculated using the 2^−ΔΔCt^ method.

### 4.11. Liver Enterobacteria Assay

After sacrificing the mice, liver tissues were excised and homogenized as previously described [5]. The liver lysate supernatants were diluted in a series and then spread onto blood agar plates for anaerobic incubation at 37 °C using Pack-Anaero equipment (Mitsubishi Gas Chemical, Tokyo, Japan).

### 4.12. Statistical Analysis

All statistical analyses were performed using the Mann–Whitney *U* test (GraphPad Prism 10.3.0). Statistical significance was set at *p* < 0.05.

## 5. Conclusions

This study analyzed the therapeutic effects of ASC-IFN-β in binge alcohol-mediated hepatic injury and gut leakiness, revealing that ASC-IFN-β secretes HGF, which prevents ethanol-induced cell death and junctional disruptions in intestinal epithelial cells. The application of a non-viral vector to overexpress *INF-β* induces HGF expression from ASCs, which might improve the clinical efficacy of engineered ASCs, thus providing a novel strategy for stem cell-induced treatment in inflammatory hepatic diseases and paving the way for further research into drug development using stem cells.

## 6. Limitations of Current Study

The current study has certain limitations. First, the efficiency of ASC delivery to target tissues was not assessed using imaging equipment. Second, the difference in the degree of treatment efficacy according to the number of cells and administration route was not evaluated. Further research is warranted to address these limitations. 

## Figures and Tables

**Figure 1 ijms-25-08509-f001:**
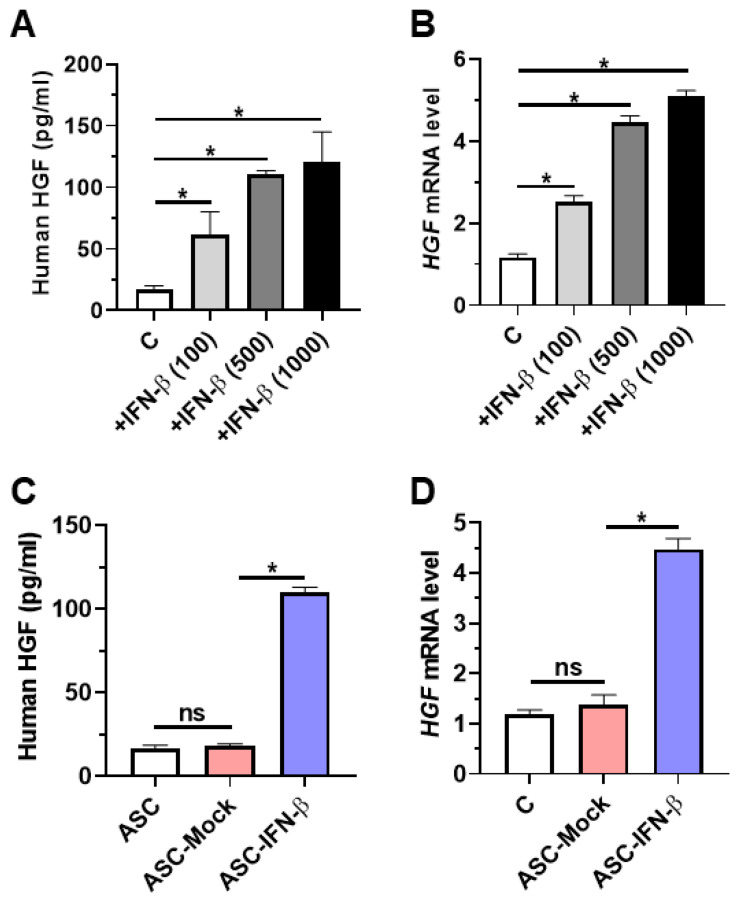
ASCs expressing IFN-β secrete HGF. Adipose tissue-derived mesenchymal stem cells (ASCs) were treated with human IFN-β (100 to 1000 units/mL) for 24 h. (**A**) ELISA of HGF in ASCs treated with human IFN-β for 24 h. (**B**) Real-time PCR analysis of HGF expression in ASCs treated with human IFN-β for 24 h. (**C**) ELISA of HGF in supernatants of ASCs transfected with a non-viral vector to carry plasmids containing the IFN-β gene for 24 h. (**D**) Real-time PCR analysis of HGF expression in ASCs transfected with a non-viral vector to carry plasmids containing the IFN-β gene for 24 h. Scatter plot. Horizontal bar; median. Significance between the treated groups was determined using the Mann–Whitney U test. * *p* < 0.05; ns—no statistical significance.

**Figure 2 ijms-25-08509-f002:**
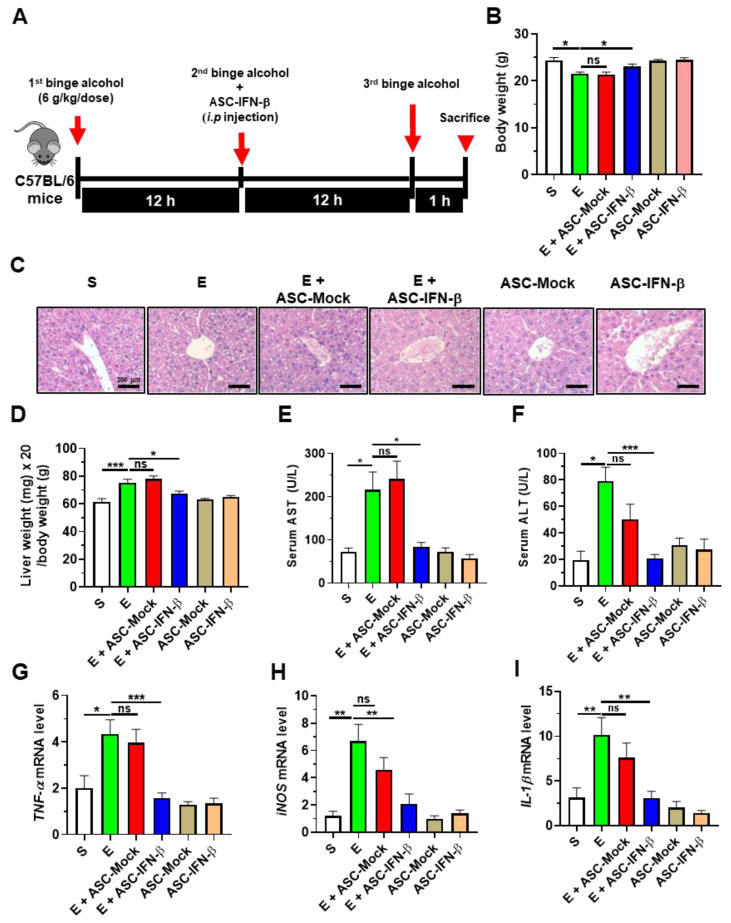
Treatment of ASCs expressing IFN-β decreases binge alcohol-induced liver damage and inflammation. C57BL/6 female mice were treated with binge ethanol. (**A**) Experimental design. C57BL/6 female mice were treated with binge alcohol three times (6 g/kg/dose). Binge alcohol-exposed C57BL/6 mice were injected intraperitoneally with ASC-IFN-β (2.0 × 10^6^ cells/200 μL of PBS) once. The mice were sacrificed 1 h after the last binge alcohol treatment. (**B**) Body weight was measured after the third injection of ethanol. (**C**) H&E staining of liver tissue. (**D**) Liver weight (mg)/body weight (g). (**E**) Serum AST levels. (**F**) Serum ALT levels (*n* = 5–17 mice per group). (**G**) TNF-α expression in the liver. (**H**) iNOS expression in the liver. (**I**) IL-1β expression in the liver. S, sham; E, binge ethanol; ASC-Mock, ASC transfected with vectors alone; ASC-IFN-β, ASC transfected with vector carrying plasmid containing the IFN-β gene. Significance between the treated groups was determined using the Mann–Whitney U test. * *p* < 0.05; ** *p* < 0.01; *** *p* < 0.001; ns—no statistical significance.

**Figure 3 ijms-25-08509-f003:**
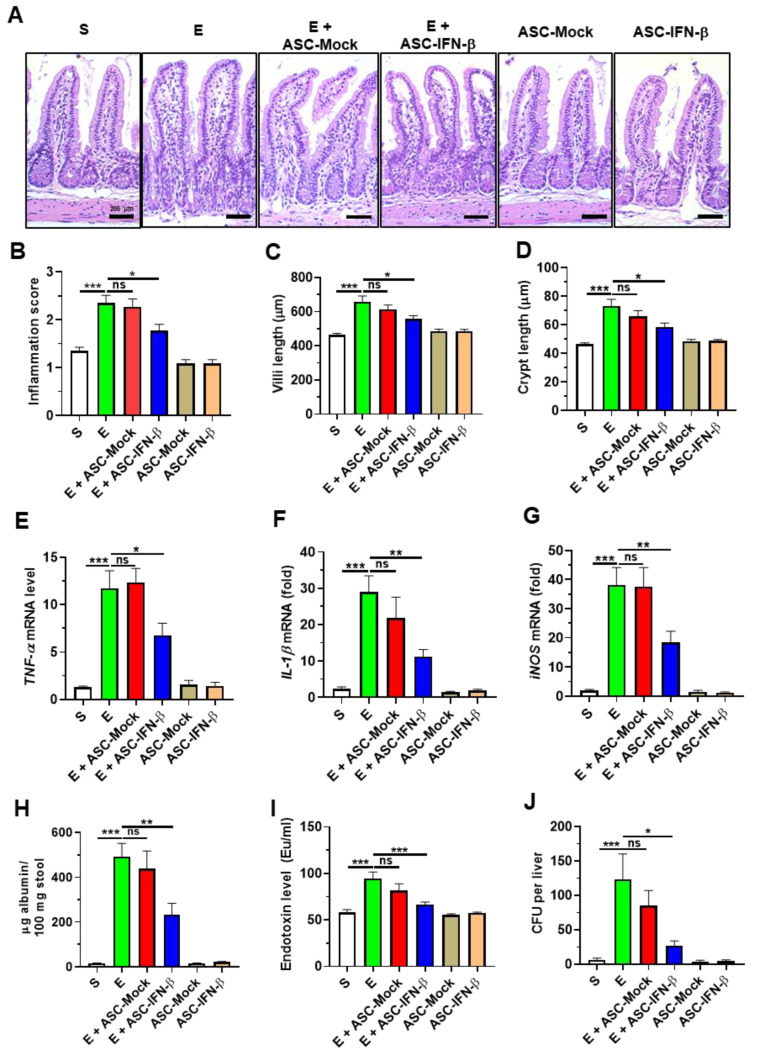
Treatment of ASCs expressing *IFN-β* decreases inflammation and barrier permeability in the small intestine. (**A**) H&E staining of the small intestine (ileum). (**B**) Inflammation scores. (**C**) Villus lengths. (**D**) Crypt lengths (*n* = 5–13 mice per group). (**E**) TNF-α expression in ileum. (**F**) IL-1β expression in ileum. (**G**) iNOS expression in ileum. (**H**) Fecal albumin levels in stool (*n* = 5–12 mice per group). (**I**) Endotoxin levels in serum (*n* = 5–12 mice per group). (**J**) Median number of colonies. S, sham; E, binge ethanol; ASC-Mock, ASCs transfected with vectors alone; ASC-IFN-β, ASCs transfected with vector carrying plasmid containing the IFN-β gene. Significance between the treated groups was determined using the Mann–Whitney U test. * *p* < 0.05; ** *p* < 0.01; *** *p* < 0.001; ns—no statistical significance.

**Figure 4 ijms-25-08509-f004:**
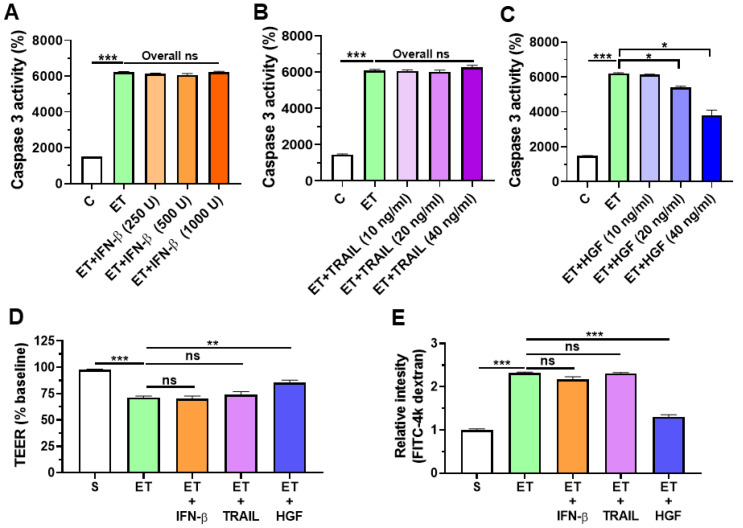
Evaluation of caspase 3 activity and permeability in ethanol-exposed intestinal epithelial cells. Caco-2 cells were treated with IFN-β, TRAIL, and HGF for 24 h, individually. Caspase 3 activity was measured via a caspase 3 commercial kit. The barrier function of Caco-2 cells was measured for transepithelial electrical resistance (TEER) and translocated FITC-4k dextran. (**A**) Caspase 3 activity (*n* = 4/group) of ethanol-exposed Caco-2 treated with IFN-β (250 to 1000 unit/mL). (**B**) Caspase 3 activity (*n* = 4/group) of ethanol-exposed Caco-2 treated with TRAIL (10 to 40 ng/mL). (**C**) Caspase 3 activity (*n* = 4/group) of ethanol-exposed Caco-2 treated with HGF (10 to 40 ng/mL). (**D**) TEER of Caco-2 treated with IFN-β, TRAIL, and HGF for 24 h, individually. (**E**) Relative intensities of FITC-4k dextran in Transwell seeded with Caco-2 monolayers treated with IFN-β (1000 unit/mL), TRAIL (40 ng/mL), or HGF (40 ng/mL) for 24 h, individually. S, sham; ET, ethanol. Significance between the treated groups was determined using the Mann–Whitney U test. * *p* < 0.05; ** *p* < 0.01; *** *p* < 0.001; ns—no statistical significance.

## Data Availability

Data are contained within the article.

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
