# Peer review of "IFN-β Overexpressing Adipose-Derived Mesenchymal Stem Cells Mitigate Alcohol-Induced Liver Damage and Gut Permeability"

_ijms, 2024, doi:10.3390/ijms25158509_

Round 1
Reviewer 1 Report
Comments and Suggestions for Authors
The article discusses a study on alcoholic liver disease (ALD), a form of hepatic inflammation caused by gut leakiness. The study evaluates the anti-inflammatory effects of adipose-derived stem cells (ASCs) overexpressing interferon-beta (IFN-β) on binge alcohol-induced liver injury and intestinal permeability. The results indicate that treatment with ASC-IFN-β reduces liver injury and inflammation and improves gut integrity, primarily through the secretion of hepatocyte growth factor (HGF). The article is thoughtful and well-organized. My only comment concerns the methodology. How ASC cells were isolated and cultured was not described in detail. There are also no tests to confirm that the cells used meet the minimum criteria for qualifying as MSCs, i.e., a profile of surface antigens and images confirming the three-lineage differentiation of the tested cells. This should be completed in the paper.
Author Response
Reviewer 1 general comment
Comments and Suggestions for Authors
The article discusses a study on alcoholic liver disease (ALD), a form of hepatic inflammation caused by gut leakiness. The study evaluates the anti-inflammatory effects of adipose-derived stem cells (ASCs) overexpressing interferon-beta (IFN-β) on binge alcohol-induced liver injury and intestinal permeability. The results indicate that treatment with ASC-IFN-β reduces liver injury and inflammation and improves gut integrity, primarily through the secretion of hepatocyte growth factor (HGF). The article is thoughtful and well-organized.
Reviewer 1, comment 1.
My only comment concerns the methodology. How ASC cells were isolated and cultured was not described in detail.
Response to Reviewer 1, comment 1.
First of all, thank you for your thorough feedback. How mesenchymal stem cells were isolated from tissues and cultured was described in detail in the methods section as follows:
Adipose tissue samples were extensively washed with Hank's Balanced Salt Solution (HBSS) to remove any unwanted materials. They were then treated with a dilute collagenase solution (0.025% in HBSS) for an hour at 37°C under gentle agitation. The remaining debris from the adipose tissue was removed by filtration through a 100-μm mesh filter. 5 million mononuclear cells were seeded in 100-mm culture dishes containing DMEM supplemented with 10% fetal bovine serum and antibiotics. After 2 days, the culture medium was changed to remove any non-adherent cells. The cell culture medium was then changed twice weekly, and the cells were passaged using trypsin/EDTA once they reached 80% confluency. For experimental studies, the cells were seeded at a density of 40,000 cells per square centimeter and cultured for the desired duration. The conditioned medium was collected, filtered, and stored at −80°C for future use. The adipose-derived stem cells (ASCs) were maintained at 37°C and 5% CO2, with the culture medium replaced every 3-4 days. When the cells reached 70-80% confluency, they were detached using trypsin and passaged.
Reviewer 1, comment 2.
There are also no tests to confirm that the cells used meet the minimum criteria for qualifying as MSCs, i.e., a profile of surface antigens and images confirming the three-lineage differentiation of the tested cells. This should be completed in the paper.
Response to Reviewer 1, comment 2.
Thank you for your detailed points. It was our mistake not to mention or cite the results of the MSC differentiation potential and marker analysis we previously used in our other papers.
Before conducting this experiment, we consulted the Institutional Review Board of Yonsei University Wonju College of Medicine (CR319098) and the Institutional Biosafety Committee (19-6 and 19-7) of Yonsei University Wonju College of Medicine to obtain the necessary approvals. When initially cultivating mesenchymal stem cell (MSC) corresponding to the above approval number, the differentiation potential and surface antigens of MSC markers had already been analyzed and confirmed, and the results for surface antigens were also included and reported in prior research (International Journal of Medical Sciences, 2020; 17(5): 609-619. doi: 10.7150/ijms.41354; https://www.medsci.org/v17p0609.htm) conducted at the same time. Since the differentiation potential and surface antigen results were presented in a figure in a research paper containing cell experiments corresponding to the same approval number above, the above paper was additionally cited in this paper.

Reviewer 2 Report
Comments and Suggestions for Authors
The article "IFN-β Overexpressing Adipose-Derived Mesenchymal Stem Cells Mitigate Alcohol-Induced Liver Damage and Gut Permeability" discusses a current topic related to Alcohol-Induced Liver Damage. Recommendations:
1. The introduction should be more concise and present the information in a better-structured manner.
2. After the introduction, the materials and methods section should follow.
3. Citations are not permitted in the results section.
4. In the discussion section, include information related to the importance of fecal microbiota transplantation in liver pathology - https://doi.org/10.3390/biomedicines11112930.
5. The limitations of the study should be placed in a separate section.
6. Line 258 – Rephrase; the phrase "in conclusion" should appear only in the final chapter.
Author Response
Reviewer 2 general comment
Comments and Suggestions for Authors
The article "IFN-β Overexpressing Adipose-Derived Mesenchymal Stem Cells Mitigate Alcohol-Induced Liver Damage and Gut Permeability" discusses a current topic related to Alcohol-Induced Liver Damage.
Reviewer 2, comment 1.
- The introduction should be more concise and present the information in a better-structured manner.
Response to Reviewer 2, comment 1.
Thank you for your constructive feedback. We strongly agree with your opinion. Reflecting on this, we reviewed the structure of the introduction again, removed unnecessary parts, and modified it to a simpler structure as follows:
- Introduction
Alcoholic liver disease (ALD) is one of the most prevalent liver diseases worldwide [1]. Excessive alcohol consumption can result in liver cirrhosis attributed to fatty liver and steatohepatitis. Approximately 50% of cirrhosis-related deaths are associated with alcohol-induced liver disease [2]. Excessive alcohol consumption can also cause significant damage to the other major organs including gut leakiness [3], contributing to increased endotoxin levels and inflammatory tissue damage to the liver [4]. In non-clinical studies, alcohol-induced liver damage mediates intestinal damage, thereby inducing intestinal inflammation and leakiness [5]. Intestinal bacteria or bacterial products translocate into the liver via a leaky barrier [5]. Adipose-derived mesenchymal stem cells (ASC) are easy to obtain clinically [6] and exhibit therapeutic effects in several inflammatory disease models [6-9], including hepatitis mouse models. Thus, research has been conducted in the direction of enhancing the therapeutic efficacy of ASC [10, 11]. Reinforcement of the function of ASC using a viral vector shows better therapeutic efficacy than conventional ASC in animal models with inflammatory diseases [12-14]. However, engineered ASC by viral-vector-mediated transduction is challenging to be used clinically or officially approved as a drug due to the risk of viral vector-induced immune response [15, 16], which promoted the other approach to evade the immune response in vivo.
A previous study demonstrated that cultivating high-density human ASC induces interferon-beta (IFN-b secretion [17, 18]. Secreted IFN-b promotes tumor necrosis factor-related apoptosis-inducing ligand (TRAIL) production in human ASC [17]. TRAIL induces antitumor effects via cell cycle arrest or apoptosis based on the type of tumor cells [17-19]. Moreover, IFN-b treatment reportedly reduces the expression of interleukin (IL)-1b and tumor necrosis factor (TNF)-a [20, 21], which play a major role in the pathogenesis of ALD [22] and leaky gut [23, 24]. A recent study showed that mouse IFN-b induces mouse hepatocyte growth factor (HGF) via signal transducer and activator of transcription 1 (STAT1) pathway in mouse mesenchymal stem cells [25]. IFN-b also stimulated HGF secretion in human peripheral blood mononuclear cells [26]. HGF treatment accelerates wound healing in major organs of humans and mice [27-29]. However, no studies have used human ASCs transfected with a non-viral vector carrying the human IFN-b gene to treat ALD.
In the current study, a non-viral vector harboring the IFN-b gene was constructed and transfected into ASCs (resulting in ASC-IFN-b). This study aimed to evaluate the anti-inflammatory potential of ASC-IFN-b in a murine experimental model of binge alcohol consumption. We found that injection of ASCs-IFN-b decreased alcohol-induced liver injury by preventing gut leakiness via HGF secreted by IFN-b-treated ASC in part. Treatment with HGF suppresses alcohol-induced cell death and disruption of junctional complexes in intestinal epithelial cells in vitro. Our results indicate that overexpression of IFN-b in ASCs using a non-viral vector may further improve the therapeutic effects of ASCs on alcoholic liver diseases via HGF.
Reviewer 2, comment 2.
- After the introduction, the materials and methods section should follow.
Response to Reviewer 2, comment 2.
Thanks for pointing this out. According to IJMS' latest submission regulations, after the result section, the discussion section appears, followed by the method section. If you check the submission regulations, you will see the same thing as above(https://www.mdpi.com/journal/ijms/instructions)
Reviewer 2, comment 3.
- Citations are not permitted in the results section.
Response to Reviewer 2, comment 3.
Thanks for the feedback. Reflecting the feedback, all citations in the result section were deleted.
Reviewer 2, comment 4.
- In the discussion section, include information related to the importance of fecal microbiota transplantation in liver pathology - https://doi.org/10.3390/biomedicines11112930.
Response to Reviewer 2, comment 4.
Here is the edited version of the academic paper in English:
The discussion section has been updated to include information about the importance of FMT in refractory liver disease, as follows:
It is challenging for a therapeutic approach using ASCs with enhanced functions to produce the same recovery effect as a liver transplant in a healthy individual. Accordingly, in addition to the functionally enhanced ASCs, Fecal Microbiota Transplantation (FMT) may be considered as a therapeutic alternative for patients with irreversible alcoholic cirrhosis. FMT has demonstrated benefits in enhancing resistance to microbial infections and improving metabolic syndrome indicators in clinical studies [41]. Given the confirmed beneficial effects of FMT [41], it is believed that this approach can be applied not only to alcoholic liver disease but also to other intestinal diseases via combined administration with functionally enhanced ASCs. Future clinical studies evaluating the effects of this combination therapy are warranted.
Reviewer 2, comment 5.
- The limitations of the study should be placed in a separate section.
Response to Reviewer 2, comment 5.
Thank you for your constructive comments. The limitations of the above paper are presented separately after the conclusion section as follows:
- Limitations of current study
The current study has certain limitations. First, the efficiency of ASC delivery to target tissues was not assessed using imaging equipment. Second, the difference in the degree of treatment efficacy according to the number of cells and administration route was not evaluated. Further research is warranted to address these limitations.
Reviewer 2, comment 6.
- Line 258 – Rephrase; the phrase "in conclusion" should appear only in the final chapter.
Response to Reviewer 2, comment 6.
Thank you for your advice on a better manuscript. The phrase "in conclusion" was removed and replaced with the phrase "in other words" instead as follows:
In other words, our study demonstrated that a non-viral vector engineered ASC-IFN-b was capable of inducing HGF secretion, which decreased ethanol-induced liver injury and gut leakiness by suppressing epithelial barrier permeability.

Round 2
Reviewer 2 Report
Comments and Suggestions for Authors
The authors have made all the recommended changes.
Author Response
Thank you for your positive comments. Thanks to this, I was able to improve the quality of the manuscript. We will revise the manuscript well to reflect other additions. Thank you.